# Causal Dependency-Aware Unsupervised Routing for Large Reasoning Models

**Jiacheng Liu** [1] **Hao Liu** [2] **Xiaofeng Hou** [3] **Wei Xue** [1] **Yike Guo** [1]

## Abstract

As Large Language Model (LLM) ecosystems grow, routing queries to the most suitable model in a diverse pool has become a critical strategy for building efficient and high-performing AI systems. A common approach is to train a supervised router; however, this requires vast, expensive human-annotated preference data and creates models that are notoriously brittle, failing to generalize when faced with inevitable distribution shifts in user queries. Consequently, developing robust, unsupervised routing methods that adapt without retraining is a crucial research frontier. This challenge is severely amplified by Large Reasoning Models (LRMs), which introduce a dual problem for any label-free method: their outputs have a causal "thinking"→"answer" structure that must be modeled, and a structural imbalance where long reasoning text can dominate the final answer signal. We introduce ReasoningRouter, a novel framework that resolves these issues with a length-balanced embedding strategy and a probabilistic model capturing the thinking-to-answer dependency. The proposed Causal Triangulation Property enables the label-free estimation of component qualities and their causal link. Beyond competitive routing accuracy, ReasoningRouter offers *unprecedented insights into model behavior*, enabling separate quality assessment of reasoning and answer components while maintaining computational efficiency.

## 1. Introduction

The landscape of artificial intelligence is increasingly characterized by a diverse ecosystem of Large Language Models

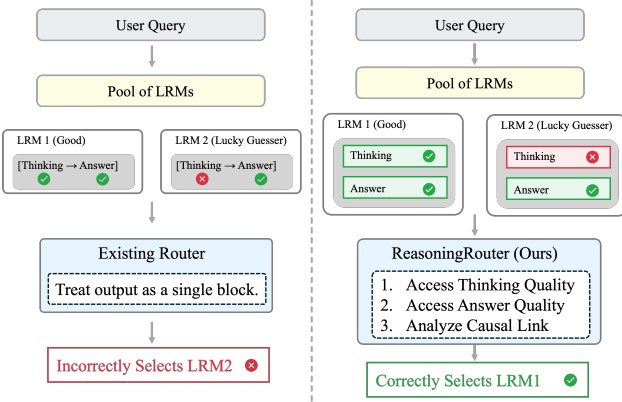

*Figure 1.* A comparison of routing methodologies for LRMs. (Left) Existing routers treat model outputs as monolithic blocks, making them incapable of distinguishing a model with sound reasoning from a "lucky guesser" that produces a correct answer from flawed logic. (Right) ReasoningRouter explicitly models the internal causal structure of LRM outputs, enabling it to select the model with the most coherent and reliable reasoning process.

(LLMs), each with unique capabilities, computational costs, and latency profiles. In this environment, deploying a single, monolithic model is often suboptimal. A more sophisticated strategy is model routing, which dynamically selects the most appropriate model from a pool of candidates for each incoming query (Guha et al., 2024). Effective routing is critical for building production systems that are not only high-performing but also efficient, intelligently allocating expensive computational resources only when a query's complexity demands it.

A common paradigm for building such routers is through supervised training. In this approach, a router model learns a mapping from a query to the best-performing LLM based on a training set of labeled examples. However, this paradigm suffers from two fundamental flaws that limit its practical utility. First, it requires the creation of vast, expensive *human-annotated preference datasets*. This process involves collecting outputs from multiple models for thousands of queries and having human annotators score their quality—a bottleneck that is costly, slow, and difficult to scale (Hu et al., 2024). Second, and more critically, routers trained this way are notoriously *brittle to distribution shift*. A model trained on a specific data distribution (e.g., academic questions)

[1]Hong Kong University of Science and Technology, Hong Kong, China [2]East China Normal University, Shanghai, China [3]Shanghai Jiao Tong University, Shanghai, China. Correspondence to: Xiaofeng Hou <hou-xf@cs.sjtu.edu.cn>, Wei Xue <weixue@ust.hk>, Yike Guo <yikeguo@ust.hk>.

*Proceedings of the 43rd International Conference on Machine Learning*, Seoul, South Korea. PMLR 306, 2026. Copyright 2026 by the author(s).

is not guaranteed to perform well when faced with a new, out-of-distribution domain (e.g., casual conversation) or as user query patterns naturally evolve over time. This reliance on a static training set makes supervised routers fragile and unreliable for dynamic, real-world applications.

The prohibitive cost and inherent brittleness of supervised methods create a compelling need for robust unsupervised routing methods that can adapt to new data without retraining. This research frontier is of paramount importance for the future of deployable AI. The challenge, however, is severely amplified by the advent of Large Reasoning Models (LRMs)—a new class of models, such as OpenAI's o1 (AI., 2024) and Google's Gemini (Comanici et al., 2025), designed to "think before they speak". These models generate outputs containing an explicit reasoning process (the "thinking") followed by a final "answer", a structure that has proven effective for complex tasks (Wei et al., 2022). While powerful, this structure introduces a dual challenge that current unsupervised methods are not equipped to handle.

First, there is a deep *causal challenge*. The "thinking" and "answer" components are not independent; a valid answer is causally dependent on a sound reasoning process, a relationship we can denote as $t_i(x) \rightarrow a_i(x)$[1]. Existing unsupervised methods, which typically treat the entire output as a single unit, ignore this dependency. This is a critical failure, as it makes them incapable of distinguishing a model that reasons its way to a correct answer from a "lucky guesser" that produces a correct answer from flawed logic (Figure 1). For trustworthy AI, rewarding the former is essential.

Second, there is a fundamental *structural challenge*. LRM outputs are textually heterogeneous. The "thinking" component can be verbose, spanning thousands of tokens of text, while the "answer" may be a single word or number. When standard embedding techniques are applied to this concatenated output, the vector representation becomes overwhelmingly dominated by the lengthy thinking part. This structural imbalance effectively masks the quality signal of the final answer, corrupting the model's overall quality assessment.

We address these fundamental challenges by introducing ReasoningRouter, a novel probabilistic framework that explicitly models and leverages the causal dependency structure inherent in reasoning models. Our approach represents a significant departure from independence-based routing by treating the thinking-to-answer relationship as a first-class modeling consideration. First, we develop a probabilistic graphical model (PGM) that captures the temporal causal structure of reasoning generation while maintaining computational tractability through carefully designed parameteri-

---

[1]We use the term "causal" in an operational sense throughout the paper, referring to a model-based directional coupling between reasoning and answer rather than an interventional causal effect.

zation. Our model explicitly represents the causal influence of thinking quality on answer quality. Second, we derive a novel causal triangulation property that can handle causal dependencies. This property has the desirable properties of closed-form parameter estimation while accounting for the correlation structure between reasoning components. Third, we introduce dependency-corrected parameter estimators that provide separate quality assessments for thinking and answer components while accounting for their causal relationship. These estimators enable fine-grained routing decisions that can prioritize different aspects of model performance depending on application requirements. Fourth, we develop a theoretically grounded coherence measure that rewards models exhibiting strong causal relationships backed by high-quality components, preventing the selection of models with spurious correlations.

The practical implications of our work extend beyond improved routing accuracy. By explicitly modeling causal dependencies, ReasoningRouter enables new capabilities such as interpretable routing decisions that explain why particular models were selected, and quality assessment that separately evaluates reasoning and answer components. These capabilities are increasingly important as reasoning models become more sophisticated and their deployment in critical applications requires greater transparency and reliability. Our contributions can be summarized as follows:

- To the best of our knowledge, we developed the first probabilistic graphical model for reasoning model routing that explicitly captures causal dependencies between thinking and answer components.

- We derive the Causal Triangulation Property that enabling the principled, label-free estimation of quality for causally dependent, structured outputs.

- We introduce a principled coherence measure that rewards models with strong causal relationships backed by high-quality components, preventing selection based on spurious correlations.

- Through experiments on diverse reasoning datasets, we demonstrate ReasoningRouter achieves superior routing accuracy and provides key insights into model behavior.

## 2. Related Work

### 2.1. Large Reasoning Model

Recent advances in LRMs have demonstrated that encouraging models to "think" longer at test time can significantly boost reasoning accuracy (Plaat et al., 2024). This approach enables LLMs to mimic complex human reasoning processes, such as tree search and reflective thinking (Xu et al.,

2025).

However, not all reasoning processes are equally effective. Reasoning can vary in quality, with different strategies producing varying levels of correctness and coherence (Feng et al., 2023; Ling et al., 2023). In general, higher-quality reasoning leads to better performance and more reliable outcomes (Qin et al., 2024; Ma et al., 2025). Nevertheless, reasoning paths tend to be dominated by long sequences of reasoning steps, which constitute the majority of the overall process (Jin et al., 2024; Sui et al., 2025).

## 2.2. Routing Strategies for LLMs

Existing routing strategies for LLMs can be broadly categorized into two types. The first category includes supervised methods, which are designed for tasks with labeled data. These approaches typically construct a performance profile (Chen et al., 2025; Pan et al., 2025; Zhang et al., 2026) or embedding representation (Zhuang et al., 2025) for each model based on its behavior on labeled datasets, and use this information to guide routing decisions for new inputs. While effective in controlled settings, such methods heavily depend on task-specific labels and often struggle to generalize to unseen or more open-ended reasoning scenarios. The second category is unsupervised methods, which do not require labeled data. One approach uses a reward model to estimate output quality and route accordingly (Lu et al., 2023), but it typically relies on another LLM to make judgments, introducing additional complexity and potential biases. Another approach scores outputs by comparing embeddings from different models, allowing routing without any supervision (Guha et al., 2024). However, it treats the entire output as a single unit, without separating the reasoning process from the final answer, making it less suitable for long reasoning tasks.

# 3. Methodology

## 3.1. Problem Formulation and Motivation

We address the fundamental challenge of routing samples to language models that produce sequential reasoning outputs consisting of a thinking component followed by an answer component. Let $\mathcal{G} = \{g_1, \ldots, g_m\}$ denote a pool of $m$ language models, where each model $g_i$ generates a structured output $(t_i(x), a_i(x))$ for input $x$, with $t_i(x)$ representing the thinking process and $a_i(x)$ the final answer.

The core challenge distinguishing reasoning model routing from traditional model selection lies in the presence of *temporal causal dependencies* that fundamentally violate the conditional independence assumptions underlying existing probabilistic routing frameworks. Specifically, in the autoregressive generation process of reasoning models, the answer $a_i(x)$ is explicitly conditioned on and causally deter-

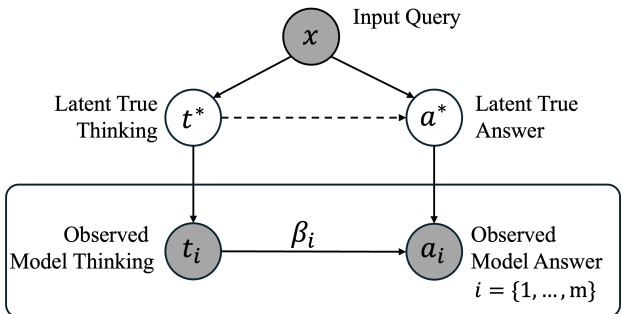

*Figure 2.* The ReasoningRouter Probabilistic Graphical Model. The model captures the generative process for a pool of $m$ reasoning models. The input query $x$ informs the latent true reasoning ($t^*$) and answer ($a^*$), which are unobserved. Each model $i$ in the pool generates an observed thinking component ($t_i$) and an answer component ($a_i$). Our model's key innovation is the explicit modeling of the causal dependency from thinking to the answer within each model, represented by the bold arrow and governed by the causal strength parameter $\beta_i$. This structure allows us to distinguish coherent reasoning from spurious correlations, a capability lacking in existing methods.

mined by the reasoning conclusions drawn in the thinking process $t_i(x)$. This creates a directed causal relationship $t_i(x) \rightarrow a_i(x)$ that existing independence-based routing methods systematically ignore.

While autoregressive language models exhibit token-level dependencies throughout generation, our focus on semantic-level causal dependencies between reasoning components is motivated by several factors. Token-level dependencies primarily capture local linguistic patterns such as grammatical coherence and syntactic structure, whereas semantic-level causal dependencies capture the global logical flow that determines reasoning quality. For routing decisions, the critical factor is whether the final answer logically follows from the reasoning process through valid inference steps, which requires modeling causal relationships between conceptual components rather than individual tokens.

Given an unlabeled test dataset $\mathcal{D}_{test} = \{x_j\}_{j=1}^n$, our objective is to construct a router route : $\mathcal{G}^m \times \mathcal{X} \rightarrow \mathcal{G}$ that selects the optimal model for each sample without access to ground truth labels, while properly accounting for the temporal causal structure inherent in reasoning generation.

## 3.2. Causal Dependency PGM

We develop a novel probabilistic graphical model that explicitly represents the one-way causal dependency from thinking to answer components, fundamentally departing from independence-based routing approaches by incorporating the temporal causal structure of reasoning generation.

**Graphical Model Structure and Causal Representation.** Figure 2 illustrates our probabilistic graphical model, where

the key innovation lies in modeling explicit causal dependencies from thinking to answer components within each model, represented by directed edges from $T_i$ to $A_i$.

This graphical representation captures the fundamental insight that thinking causally determines the answer within each reasoning model through the temporal generation process, while maintaining conditional independence between different models given the latent truth. The directed arrows represent this causal structure, distinguishing our approach from both independence-based methods that ignore dependencies and symmetric correlation approaches that fail to capture the temporal asymmetry.

**Length-Balanced Embedding Strategy.** A fundamental implementation challenge in reasoning model evaluation stems from the significant length and semantic density disparities between thinking and answer components. Thinking components typically contain numerous of tokens with detailed reasoning steps, mathematical derivations, and intermediate conclusions, while answers are often short factual statements. This length imbalance creates several problems for embedding-based quality assessment.

First, naive concatenation of thinking and answer components results in embeddings dominated by the lengthy thinking content, effectively hiding answer-specific information. Second, separate embedding of raw components loses the contextual relationship between reasoning and conclusions. Third, the semantic density varies significantly, thinking components contain redundant explanatory text while answers contain concentrated factual content.

We address these challenges through a length-balanced embedding strategy that preserves both component-specific quality information and their contextual relationship:

$$\boldsymbol{\lambda}_i^T(x) = z_{g_0}([x, t_i(x)]) \in \mathbb{R}^d \tag{1}$$
$$\boldsymbol{\lambda}_i^A(x) = z_{g_0}([x, a_i(x)]) \in \mathbb{R}^d \tag{2}$$

where $z_{g_0}(\cdot)$ denotes the embedding function of a frozen pretrained encoder $g_0$, with both components mapped to the same dimension $d$ to enable meaningful causal dependency computation. The crucial insight is that by embedding each component with the input context $x$, we maintain semantic grounding while avoiding length imbalance issues that would arise from concatenating thinking and answer directly.

We define corresponding latent true embeddings that represent the optimal reasoning process and answer:

$$\mathbf{z}^{*T}(x) = z_{g_0}([x, t^*(x)]) \tag{3}$$
$$\mathbf{z}^{*A}(x) = z_{g_0}([x, a^*(x)]) \tag{4}$$

where $t^*(x)$ and $a^*(x)$ represent the optimal reasoning process and answer, which are unobserved but can be estimated through our causal triangulation approach.

**Causal Dependency Modeling.** To model the causal dependency from thinking to answer components through a conditional probability formulation that respects the temporal generation order while maintaining computational tractability. We formulate the problem using conditional distributions that directly capture the causal structure:

$$\boldsymbol{\lambda}_i^T(x) - \mathbf{z}^{*T}(x) \sim \mathcal{N}\left(\mathbf{0}, \frac{1}{2\theta_i^T(x)}\mathbf{I}_d\right) \tag{5}$$

$$\boldsymbol{\lambda}_i^A(x) - \mathbf{z}^{*A}(x) \mid \boldsymbol{\lambda}_i^T(x) \sim \mathcal{N}\left(\boldsymbol{\mu}_i^{A|T}(x), \frac{1}{2\theta_i^{A|T}(x)}\mathbf{I}_d\right) \tag{6}$$

where the conditional mean captures the causal dependency:

$$\boldsymbol{\mu}_i^{A|T}(x) = \beta_i(x)(\boldsymbol{\lambda}_i^T(x) - \mathbf{z}^{*T}(x)) \tag{7}$$

Here, $\beta_i(x) \in \mathbb{R}$ is a scalar causal influence parameter that measures how deviations in thinking quality propagate to answer quality.

**Joint Distribution and Marginal Covariance Derivation.** While the causal structure is naturally expressed through conditional distributions, our triangulation-based parameter estimation requires the marginal joint distribution. We derive the marginal covariance structure through careful integration of the conditional formulation.

The marginal joint distribution of $(\boldsymbol{\lambda}_i^T, \boldsymbol{\lambda}_i^A)$ has the block-structured covariance matrix:

$$\boldsymbol{\Sigma}_i(x) = \begin{bmatrix} \frac{1}{2\theta_i^T(x)}\mathbf{I}_d & \frac{\beta_i(x)}{2\theta_i^T(x)}\mathbf{I}_d \\ \frac{\beta_i(x)}{2\theta_i^T(x)}\mathbf{I}_d & \left(\frac{1}{2\theta_i^{A|T}(x)} + \frac{\beta_i^2(x)}{2\theta_i^T(x)}\right)\mathbf{I}_d \end{bmatrix} \tag{8}$$

This marginal covariance structure reveals several important properties. This covariance structure encodes a unidirectional causal mechanism: the off-diagonal terms quantify the covariance induced by the propagation of thinking uncertainty to answer uncertainty, rather than a symmetric correlation. The marginal answer variance decomposes into intrinsic answer uncertainty plus propagated thinking uncertainty, providing insight into the sources of answer quality variation.

We also define the marginal answer precision for notational convenience:

$$\theta_i^A(x) = \frac{\theta_i^{A|T}(x)\theta_i^T(x)}{\theta_i^T(x) + \beta_i^2(x)\theta_i^{A|T}(x)} \tag{9}$$

This represents the effective precision of the answer component, accounting for both its intrinsic quality $\theta_i^{A|T}(x)$ and the propagated uncertainty from thinking through the causal parameter $\beta_i(x)$.

### 3.3. Causal Parameter Estimation

The key theoretical challenge is developing triangulation equations that account for causal structure without losing the fundamental properties that enable tractable estimation.

**Causal Triangulation Property.** The foundation of our parameter estimation approach is a novel triangulation property that accounts for causal dependencies while preserving the distance-based triangulation structure that enables closed-form parameter estimation.

**Theorem 3.1** (Causal Triangulation Property). *For the causal probabilistic model defined in Eqs.* (5)-(6), *the outputs of any two distinct models $i$ and $j$ are conditionally independent given the latent true embeddings $\mathbf{z}^{*T}$ and $\mathbf{z}^{*A}$. Define the weighted distance to the latent truth for model $i$ as:*

$$\delta_{i*}^{weighted}(x) = \alpha E[\|\boldsymbol{\lambda}_i^T - \mathbf{z}^{*T}\|^2] + (1-\alpha)E[\|\boldsymbol{\lambda}_i^A - \mathbf{z}^{*A}\|^2] \quad (10)$$

*and the weighted distance between models $i$ and $j$ as:*

$$\delta_{ij}^{weighted}(x) = \alpha E[\|\boldsymbol{\lambda}_i^T - \boldsymbol{\lambda}_j^T\|^2] + (1-\alpha)E[\|\boldsymbol{\lambda}_i^A - \boldsymbol{\lambda}_j^A\|^2] \quad (11)$$

*where $\alpha \in [0,1]$. Then the standard triangulation property holds:*

$$\delta_{ij}^{weighted}(x) = \delta_{i*}^{weighted}(x) + \delta_{j*}^{weighted}(x) \quad (12)$$

The detailed proof is provided in the appendix.

### 3.3.1. CAUSAL PARAMETER ESTIMATION ALGORITHMS

For any triple of models $(i, j, k)$, the triangulation property (12) gives us:

$$\delta_{i*}^{weigh}(x) = \frac{1}{2}\left(\delta_{ij}^{weigh}(x) + \delta_{ik}^{weigh}(x) - \delta_{jk}^{weigh}(x)\right) \quad (13)$$

Let $\hat{\delta}_{ij}^{(\alpha)}$ be the empirical estimate of the weighted distance from data. Then, we define $\hat{D}_{i,jk}^{(\alpha)} = \frac{1}{2}\left(\hat{\delta}_{ij}^{(\alpha)} + \hat{\delta}_{ik}^{(\alpha)} - \hat{\delta}_{jk}^{(\alpha)}\right)$. We have:

$$\hat{D}_{i,jk}^{(\alpha)} \approx \delta_{i*}^{weigh}(x) = \alpha \frac{d}{2\theta_i^T} + (1-\alpha)\frac{d}{2\theta_i^A} \quad (14)$$

To separate $\theta_i^T$ and $\theta_i^A$, we use two different weight values, e.g., $\alpha_1 = 0.8$ and $\alpha_2 = 0.2$ (the choice of these values is a mathematical tool for resolving the system; it is not

a hyperparameter, and any pair of distinct values for $\alpha \in (0, 1)$ will yield the same solution.), to create a system of two linear equations:

$$\hat{D}_{i,jk}^{(0.8)} = 0.8\frac{d}{2\theta_i^T} + 0.2\frac{d}{2\theta_i^A} \quad (15)$$

$$\hat{D}_{i,jk}^{(0.2)} = 0.2\frac{d}{2\theta_i^T} + 0.8\frac{d}{2\theta_i^A} \quad (16)$$

Solving this linear system for the precision terms $(1/\theta)$ yields:

$$\frac{d}{2\hat{\theta}_i^T(x)} = \frac{0.8\hat{D}_{i,jk}^{(0.8)} - 0.2\hat{D}_{i,jk}^{(0.2)}}{0.8^2 - 0.2^2} = \frac{4\hat{D}_{i,jk}^{(0.8)} - \hat{D}_{i,jk}^{(0.2)}}{3} \quad (17)$$

$$\frac{d}{2\hat{\theta}_i^A(x)} = \frac{0.8\hat{D}_{i,jk}^{(0.2)} - 0.2\hat{D}_{i,jk}^{(0.8)}}{0.8^2 - 0.2^2} = \frac{4\hat{D}_{i,jk}^{(0.2)} - \hat{D}_{i,jk}^{(0.8)}}{3} \quad (18)$$

This gives us direct estimators for the marginal thinking quality $\hat{\theta}_i^T$ and the marginal answer quality $\hat{\theta}_i^A$.

The estimation proceeds in three steps:

1. **Estimate Marginal Qualities:** Use the equations above to compute $\hat{\theta}_i^T(x)$ and $\hat{\theta}_i^A(x)$ for all models $i = 1, \ldots, m$. This step does not require knowledge of $\beta_i$.

2. **Estimate Latent Truth:** Compute the latent true embeddings as a quality-weighted average of the observed embeddings:

$$\hat{\mathbf{z}}^{*T}(x) = \frac{\sum_{j=1}^m \hat{\theta}_j^T(x)\boldsymbol{\lambda}_j^T(x)}{\sum_{j=1}^m \hat{\theta}_j^T(x)} \quad (19)$$

$$\hat{\mathbf{z}}^{*A}(x) = \frac{\sum_{j=1}^m \hat{\theta}_j^A(x)\boldsymbol{\lambda}_j^A(x)}{\sum_{j=1}^m \hat{\theta}_j^A(x)} \quad (20)$$

3. **Estimate Causal Strength:** Now that we have estimates for the latent truth, we can estimate the causal parameter $\beta_i(x)$ using its definition from the conditional mean. This is a simple linear regression problem. The ordinary least squares estimator is:

$$\hat{\beta}_i(x) = \frac{\langle \boldsymbol{\lambda}_i^A(x) - \hat{\mathbf{z}}^{*A}(x), \boldsymbol{\lambda}_i^T(x) - \hat{\mathbf{z}}^{*T}(x)\rangle}{\|\boldsymbol{\lambda}_i^T(x) - \hat{\mathbf{z}}^{*T}(x)\|^2} \quad (21)$$

### 3.3.2. FINAL QUALITY SCORE COMPUTATION

The final routing score can integrate the estimated parameters. A possible heuristic is:

$$\hat{\theta}_i(x) = w_T\hat{\theta}_i^T(x) + w_A\hat{\theta}_i^A(x) + w_C \cdot \text{coherence}_i(x) \quad (22)$$

where $w_T, w_A, w_C$ are weights and the coherence term rewards models where a strong causal link is backed by high-quality components:

$$\text{coherence}_i(x) = \frac{|\hat{\beta}_i(x)|}{1 + |\hat{\beta}_i(x)|} \cdot \min(\hat{\theta}_i^T(x), \hat{\theta}_i^A(x)) \quad (23)$$

This coherence formulation rewards models where strong causal relationships are supported by high quality in both components, preventing models with strong but spurious correlations from being overvalued.

Our method is *computationally efficient*, adding minimal overhead to standard output-based routing. The primary cost of embedding generation is handled by a lightweight 0.3 0.6B model, making it substantially faster than inference on the 3B-30B candidate LRMs. Crucially, our core parameter estimation is a non-iterative, closed-form calculation that avoids expensive training and adds only negligible, millisecond-level latency to the final routing decision.

# 4. Experiment

## 4.1. Experiment Setup

We evaluate ReasoningRouter on 9 benchmark datasets spanning diverse reasoning tasks: ARC (Clark et al., 2018), OpenBookQA (Open.) (Mihaylov et al., 2018) and SocialiQA (Social.) (Sap et al., 2019) for commonsense reasoning; GSM8K (Cobbe et al., 2021) and MathQA (Math.) (Amini et al., 2019) for mathematical problem solving; MedMCQA (Med.) (Pal et al., 2022) for medical knowledge; SciQ (Johannes Welbl, 2017) for science questions; TruthfulQA (Truth. ) (Lin et al., 2021) and MMLU-Pro (MP.) (Wang et al., 2024) for multi-domain understanding. Our experiments assess the effectiveness of score-based routing in selecting high-quality models for each sample across all datasets.

Our inference pool consists of eight open large reasoning models: DeepHermes-3-Llama-3-3B-Preview, DeepHermes-3-Mistral-24B-Preview (Teknium et al., 2025), OpenReasoning-Nemotron-14B, OpenReasoning-Nemotron-7B (Ahmad et al., 2025), Qwen3-14B, Qwen3-30B-A3B, Qwen3-8B (Team, 2025), SmolLM3-3B (Bakouch et al., 2025). These models vary in size (3B to 30B parameters), architecture, and training style.

To evaluate the effectiveness of our routing approach, we compare it against several other unsupervised routing baselines. The first is a random routing strategy, where each input is assigned to a model uniformly at random from the pool (RandomRouter). This serves as a simple but meaningful baseline to assess whether any structured selection mechanism provides consistent gains. We also compare against three different variants of the state-of-the-art routing

framework (Guha et al., 2024), which use different parts of the model's output embedding for routing decisions. Specifically, one variant uses only the embedding of the reasoning trace (ThinkingRouter), another uses only the answer embedding (AnswerRouter), and the third treats the entire output as a single, monolithic embedding (OutputRouter).

We use the widely used BGE-m3 (Chen et al., 2024) and Embeddinggemma-300m (Schechter Vera et al., 2025) to generate embeddings for semantic distance measurement, and we use accuracy as the primary evaluation metric, defined as the number of correct predictions divided by the total number of samples. The code is available on GitHub.

## 4.2. Performance Comparison

The routing accuracy results are reported in Table 1 under two different embedding models. Across both settings, our proposed method ReasoningRouter consistently achieves the best overall performance, demonstrating strong robustness to the choice of embedding model. When using BGE-M3, ReasoningRouter attains the highest average accuracy of 88.20% with the best average rank of 1.44. It achieves first place on five out of nine benchmarks, including GSM8K, MedMCQA, SciQ, SocialIQA, and TruthfulQA, and remains highly competitive on the remaining tasks. Similar trends are observed with Embeddinggemma-300m, where ReasoningRouter again achieves the highest average accuracy of 88.23% and the best average rank of 1.56. Across both embedding backbones, ReasoningRouter consistently places either first or second on every benchmark, never falling below second place. This consistent dominance across two distinct embedding backbones highlights the generality and stability of our routing strategy.

Both ThinkingRouter and AnswerRouter consistently outperform RandomRouter across nearly all datasets. This baseline improvement demonstrates that both the final answers and the internal reasoning traces contain meaningful signals that can effectively guide model routing decisions. However, neither method establishes a decisive lead over the other. This suggests that relying exclusively on a single information source is insufficient to capture the subtle, actionable distinctions in model quality.

The OutputRouter, which treats the entire model output as a monolithic embedding, performs well on several benchmarks due to its holistic views. However, it underperforms on many datasets like GSM8K and MathQA, suggesting limitations in how it balances different components of the response. One potential issue is length imbalance: reasoning traces are typically much longer than answers, causing the embedding space to be dominated by the "thinking" portion. As a result, subtle but important differences in answer quality or correctness may be overshadowed, reducing routing accuracy on tasks where answer precision is paramount.

| Methods | ARC | GSM8k | Math. | Med. | MP. | Open. | SciQ | Social. | Truth. | Avg. |
|---|---|---|---|---|---|---|---|---|---|---|
| RandomRouter | 92.37(5) | 93.64(5) | 93.62(5) | 69.67(5) | 78.07(5) | 90.66(5) | 95.87(5) | 77.43(5) | 73.57(5) | 84.99(5.00) |
| ThinkingRouter | **94.03(1)** | 95.23(4) | **94.78(1)** | 72.74(4) | 81.76(4) | 92.54(4) | 97.27(3) | 80.56(2) | 80.48(2) | 87.71(2.78) |
| AnswerRouter | 93.79(4) | 95.35(2) | 93.98(4) | 72.84(3) | **82.68(1)** | **92.95(1)** | 96.70(4) | 78.63(4) | 76.58(4) | 87.06(3.00) |
| OutputRouter | 93.96(3) | 95.29(3) | 94.60(2) | 73.50(2) | 82.02(3) | 92.68(3) | 97.51(2) | 80.56(2) | 80.18(3) | 87.81(2.56) |
| **ReasoningRouter** | 94.01(2) | **95.78(1)** | 94.60(2) | **74.06(1)** | 82.20(2) | 92.88(2) | **97.52(1)** | **80.74(1)** | **81.98(1)** | **88.20(1.44)** |

*(a)* Using **BGE-M3** as embedding model

| Methods | ARC | GSM8k | Math. | Med. | MP. | Open. | SciQ | Social. | Truth. | Avg. |
|---|---|---|---|---|---|---|---|---|---|---|
| RandomRouter | 92.37(5) | 93.64(5) | 93.62(3) | 69.67(5) | 78.07(5) | 90.66(5) | 95.87(5) | 77.43(5) | 73.57(5) | 84.99(4.78) |
| ThinkingRouter | 93.96(3) | 95.19(3) | 93.45(4) | 73.31(4) | **82.90(1)** | 93.02(3) | 97.07(3) | 79.51(4) | 82.28(3) | 87.85(3.11) |
| AnswerRouter | 93.89(4) | **95.93(1)** | **94.16(1)** | 73.87(3) | 82.20(4) | 92.71(4) | 96.68(4) | **80.21(1)** | 74.77(4) | 87.16(2.89) |
| OutputRouter | 94.22(2) | 95.12(4) | 93.27(5) | **75.56(1)** | 82.28(3) | 93.22(2) | **97.23(1)** | 79.68(3) | **82.88(1)** | 88.16(2.44) |
| **ReasoningRouter** | **94.27(1)** | 95.73(2) | 93.98(2) | 74.06(2) | 82.50(2) | **93.59(1)** | **97.23(1)** | 79.86(2) | **82.88(1)** | **88.23(1.56)** |

*(b)* Using **Embeddinggemma-300m** as embedding model

*Table 1.* Performance comparison across different embedding models. Numbers in parentheses indicate the rank in each task. Best results are **bolded**, second-best underlined.

In contrast, ReasoningRouter employs an integrated scoring function that explicitly models multiple dimensions of response quality: the quality of reasoning, the semantic of the answer, and the consistency between reasoning and answer. By decomposing outputs and scoring each component, ReasoningRouter enables more accurate and robust routing across diverse reasoning tasks.

### 4.3. Routing Behavior Analysis

In this section, We analyze ReasoningRouter's routing behavior to understand the sources of its performance gains. Given the consistent trends observed across various embedding configurations in our main experiments, we focus the following analysis on the BGE-M3 case as a representative instance for brevity.

#### 4.3.1. ROUTING SCORES

To evaluate the effectiveness of our routing mechanism in distinguishing high-quality models from low-quality ones, we analyze the distribution of routing scores generated by ReasoningRouter across various benchmark datasets. For each input, we compare the highest routing score among correct models with the highest routing score among incorrect models. As shown in Figure 3, there is a clear separation between these two distributions across most datasets. This indicates that the router tends to assign higher confidence to correct models than to incorrect ones, demonstrating its ability to reliably differentiate between correct and incorrect predictions.

#### 4.3.2. COHERENCE RELATIONSHIPS

We evaluate model behavior using the coherence score defined in Equation (23), which jointly captures the strength of

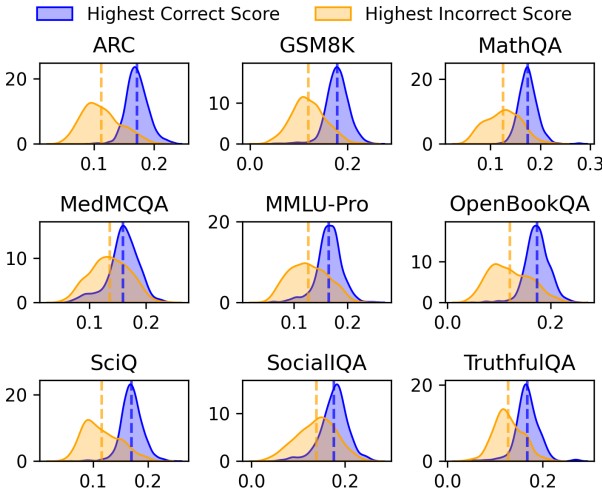

*Figure 3.* Distribution of routing scores for the highest-scoring correct and incorrect models across benchmark datasets. The dashed lines indicate the mean scores.

causal associations and the quality of supporting evidence. Figure 4 compares the coherence distributions for correct and incorrect answers across several benchmark datasets. The results show that correct responses consistently achieve higher coherence than incorrect ones across datasets, suggesting that accurate predictions are grounded in more reliable reasoning. This supports the use of coherence as a signal for output reliability.

#### 4.3.3. HYPERPARAMETER SENSITIVITY

We conduct a sensitivity analysis on the key hyperparameters $w_t$ and $w_a$, which jointly control the relative contributions of the reasoning trace score and the answer score in

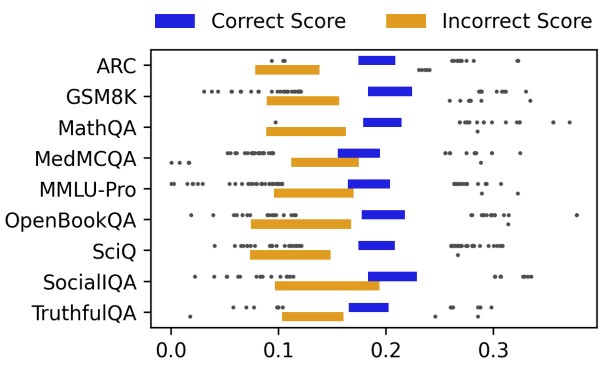

*Figure 4.* Coherence scores for correct and incorrect models across various datasets.

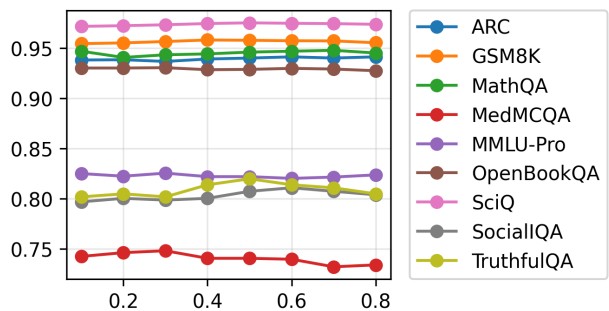

*Figure 5.* Hyperparameter sensitivity analysis of the weight. The x-axis represents the proportion of $w_T$, and the y-axis shows the accuracy across different benchmark datasets.

the routing mechanism. As illustrated in Figure 5, routing performance remains largely stable across all benchmarks over a wide range of weight proportions, indicating that the proposed method is not highly sensitive to how the two components are weighted and this consistent behavior across benchmarks highlights the robustness of the routing mechanism.

### 4.3.4. MODEL POOL SIZE

To investigate the impact of model pool size, we analyze how the number of inference models affects the performance of ReasoningRouter. For each model pool size, we average the performance over all possible orderings of the candidate models, ensuring that the analysis is not affected by ordering effects. As shown in Figure 6, the average accuracy across the nine benchmarks consistently increases as more models are included in the pool. This improvement comes from two key factors. First, a larger model pool increases the likelihood of including well-performing models for diverse input types, expanding the router's selection range. Second, with more models producing varied outputs, the router is exposed to a richer set of response characteristics, enabling it to more accurately evaluate model behavior and distinguish

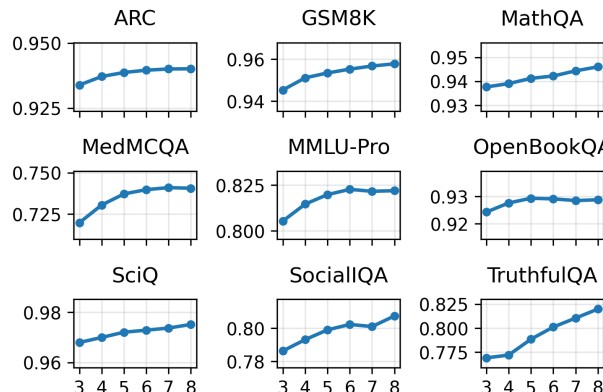

*Figure 6.* Impact of model pool size on routing performance. The x-axis shows the number of models in the candidate pool, and the y-axis reports the accuracy on each datasets.

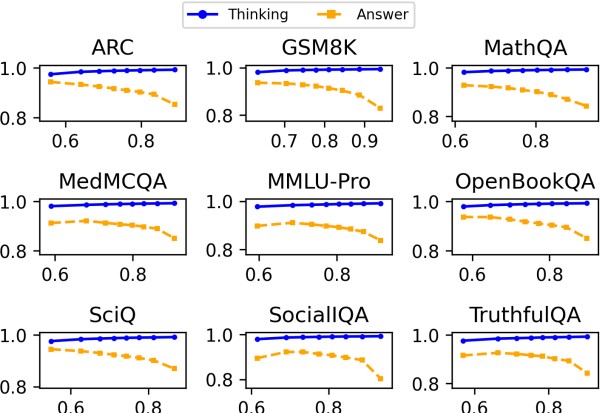

*Figure 7.* Relationship between thinking content ratio (thinking tokens / total output tokens) and embedding similarity across benchmark datasets. The x-axis shows the proportion of thinking tokens relative to the full output length. The y-axis shows cosine similarity between the full output embedding and (i) the thinking embedding (blue curves) or (ii) the answer embedding (orange curves).

high-quality responses.

### 4.3.5. OUTPUT EMBEDDING DOMINANCE VALIDATION

To empirically examine how different components of model outputs align with the overall output embedding, we conduct a systematic analysis of the embedding similarity between each component and the full output representation. Figure 7 validates that thinking content dominates the full output embedding space. As thinking ratio increases, cosine similarity to the thinking embedding remains high, while similarity to the answer embedding drops. This confirms our hypothesis: longer reasoning traces disproportionately shape monolithic embeddings, obscuring answer-specific signals.

# 5. Conclusion

We propose ReasoningRouter, a label-free routing method for Large Reasoning Models that models the causal link between reasoning and answers while balancing long reasoning texts. Leveraging the Causal Triangulation Property, it effectively estimates output quality without supervision. Experiments show our model outperforms prior methods on diverse reasoning tasks, highlighting the benefit of modeling reasoning structure for robust, adaptive routing.

# Acknowledgments

We sincerely thank all the anonymous reviewers for their valuable comments and feedback. This work is supported by the National Natural Science Foundation of China (No.62502302 & No.62441225), Theme-based Research Scheme under Grant T45-205/21-N from Hong Kong RGC, and Generative AI Research and Development Centre from InnoHK.

# Impact Statement

This paper presents work whose goal is to advance the field of machine learning. There are many potential societal consequences of our work, none of which we feel must be specifically highlighted here.

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

## A. Derivation of Eq. (8)

The derivation follows from the law of total covariance. For the marginal variance of the answer component:

$$\text{Var}(\boldsymbol{\lambda}_i^A - \mathbf{z}^{*A}) \tag{24}$$

$$= \mathbb{E}[\text{Var}(\boldsymbol{\lambda}_i^A - \mathbf{z}^{*A}|\boldsymbol{\lambda}_i^T)] + \text{Var}(\mathbb{E}[\boldsymbol{\lambda}_i^A - \mathbf{z}^{*A}|\boldsymbol{\lambda}_i^T]) \tag{25}$$

$$= \frac{1}{2\theta_i^{A|T}(x)}\mathbf{I}_d + \text{Var}(\beta_i(x)(\boldsymbol{\lambda}_i^T - \mathbf{z}^{*T})) \tag{26}$$

$$= \frac{1}{2\theta_i^{A|T}(x)}\mathbf{I}_d + \frac{\beta_i^2(x)}{2\theta_i^T(x)}\mathbf{I}_d \tag{27}$$

For the cross-covariance:

$$\text{Cov}(\boldsymbol{\lambda}_i^T - \mathbf{z}^{*T}, \boldsymbol{\lambda}_i^A - \mathbf{z}^{*A}) \tag{28}$$

$$= \mathbb{E}[(\boldsymbol{\lambda}_i^T - \mathbf{z}^{*T})(\beta_i(x)(\boldsymbol{\lambda}_i^T - \mathbf{z}^{*T}))^T] \tag{29}$$

$$= \frac{\beta_i(x)}{2\theta_i^T(x)}\mathbf{I}_d \tag{30}$$

## B. Proof of Theorem 1

**Causal Dependency Modeling.** To model the causal dependency from thinking to answer components through a conditional probability formulation that respects the temporal generation order while maintaining computational tractability. We formulate the problem using conditional distributions that directly capture the causal structure:

$$\boldsymbol{\lambda}_i^T(x) - \mathbf{z}^{*T}(x) \sim \mathcal{N}\left(\mathbf{0}, \frac{1}{2\theta_i^T(x)}\mathbf{I}_d\right) \tag{31}$$

$$\boldsymbol{\lambda}_i^A(x) - \mathbf{z}^{*A}(x) \mid \boldsymbol{\lambda}_i^T(x) \sim \mathcal{N}\left(\boldsymbol{\mu}_i^{A|T}(x), \frac{1}{2\theta_i^{A|T}(x)}\mathbf{I}_d\right) \tag{32}$$

where the conditional mean captures the causal dependency:

$$\boldsymbol{\mu}_i^{A|T}(x) = \beta_i(x)(\boldsymbol{\lambda}_i^T(x) - \mathbf{z}^{*T}(x)) \tag{33}$$

Here, $\beta_i(x) \in \mathbb{R}$ is a scalar causal influence parameter that measures how deviations in thinking quality propagate to answer quality.

**Theorem B.1** (Causal Triangulation Property). *For the causal probabilistic model defined in Eqs. (31)-(32), the outputs of any two distinct models $i$ and $j$ are conditionally independent given the latent true embeddings $\mathbf{z}^{*T}$ and $\mathbf{z}^{*A}$. Define the weighted distance to the latent truth for model $i$ as:*

$$\delta_{i*}^{weighted}(x) = \alpha E[\|\boldsymbol{\lambda}_i^T - \mathbf{z}^{*T}\|^2] + (1-\alpha)E[\|\boldsymbol{\lambda}_i^A - \mathbf{z}^{*A}\|^2] \tag{34}$$

*and the weighted distance between models $i$ and $j$ as:*

$$\delta_{ij}^{weighted}(x) = \alpha E[\|\boldsymbol{\lambda}_i^T - \boldsymbol{\lambda}_j^T\|^2] + (1-\alpha)E[\|\boldsymbol{\lambda}_i^A - \boldsymbol{\lambda}_j^A\|^2] \tag{35}$$

*where $\alpha \in [0,1]$. Then the standard triangulation property holds:*

$$\delta_{ij}^{weighted}(x) = \delta_{i*}^{weighted}(x) + \delta_{j*}^{weighted}(x) \tag{36}$$

*Proof.* The proof relies on the conditional independence between models $i$ and $j$ (for $i \neq j$) given the latent truth $\mathbf{z}^*$. Let $\varepsilon_i = (\varepsilon_i^T, \varepsilon_i^A)$ and $\varepsilon_j = (\varepsilon_j^T, \varepsilon_j^A)$ be the error vectors for models $i$ and $j$. By the model definition, $\varepsilon_i$ and $\varepsilon_j$ are independent.

First, consider the thinking components.

$$E[\|\boldsymbol{\lambda}_i^T - \boldsymbol{\lambda}_j^T\|^2] = E[\|(\boldsymbol{\lambda}_i^T - \mathbf{z}^{*T}) - (\boldsymbol{\lambda}_j^T - \mathbf{z}^{*T})\|^2] \tag{37}$$

$$= E[\|\boldsymbol{\varepsilon}_i^T - \boldsymbol{\varepsilon}_j^T\|^2] \tag{38}$$

$$= E[\|\boldsymbol{\varepsilon}_i^T\|^2] + E[\|\boldsymbol{\varepsilon}_j^T\|^2] - 2E[\boldsymbol{\varepsilon}_i^{T\top}\boldsymbol{\varepsilon}_j^T] \tag{39}$$

Since $\varepsilon_i^T$ and $\varepsilon_j^T$ are independent and have zero mean, $E[\varepsilon_i^{T\top}\varepsilon_j^T] = E[\varepsilon_i^{T\top}]E[\varepsilon_j^T] = \mathbf{0}$. The variance of $\varepsilon_i^T$ is $\frac{1}{2\theta_i^T}\mathbf{I}_d$, so its expected squared norm is $\mathrm{Tr}(\frac{1}{2\theta_i^T}\mathbf{I}_d) = \frac{d}{2\theta_i^T}$. Thus,

$$E[\|\boldsymbol{\lambda}_i^T - \boldsymbol{\lambda}_j^T\|^2] = \frac{d}{2\theta_i^T} + \frac{d}{2\theta_j^T} = E[\|\boldsymbol{\varepsilon}_i^T\|^2] + E[\|\boldsymbol{\varepsilon}_j^T\|^2] \tag{40}$$

Next, consider the answer components. The same logic applies.

$$E[\|\boldsymbol{\lambda}_i^A - \boldsymbol{\lambda}_j^A\|^2] = E[\|(\boldsymbol{\lambda}_i^A - \mathbf{z}^{*A}) - (\boldsymbol{\lambda}_j^A - \mathbf{z}^{*A})\|^2] \tag{41}$$

$$= E[\|\boldsymbol{\varepsilon}_i^A - \boldsymbol{\varepsilon}_j^A\|^2] \tag{42}$$

$$= E[\|\boldsymbol{\varepsilon}_i^A\|^2] + E[\|\boldsymbol{\varepsilon}_j^A\|^2] - 2E[\boldsymbol{\varepsilon}_i^{A\top}\boldsymbol{\varepsilon}_j^A] \tag{43}$$

Due to the conditional independence of models, $\varepsilon_i^A$ and $\varepsilon_j^A$ are also independent, so $E[\varepsilon_i^{A\top}\varepsilon_j^A] = 0$. The marginal variance of $\varepsilon_i^A$ is $\left(\frac{1}{2\theta_i^{A|T}} + \frac{\beta_i^2}{2\theta_i^T}\right)\mathbf{I}_d = \frac{1}{2\theta_i^A}\mathbf{I}_d$. Its expected squared norm is $\frac{d}{2\theta_i^A}$. Thus,

$$E[\|\boldsymbol{\lambda}_i^A - \boldsymbol{\lambda}_j^A\|^2] = \frac{d}{2\theta_i^A} + \frac{d}{2\theta_j^A} = E[\|\boldsymbol{\varepsilon}_i^A\|^2] + E[\|\boldsymbol{\varepsilon}_j^A\|^2] \tag{44}$$

Combining the components for the weighted distance:

$$\delta_{ij}^{weigh}(x) \tag{45}$$

$$= \alpha\left(\frac{d}{2\theta_i^T} + \frac{d}{2\theta_j^T}\right) + (1-\alpha)\left(\frac{d}{2\theta_i^A} + \frac{d}{2\theta_j^A}\right) \tag{46}$$

$$= \left(\alpha\frac{d}{2\theta_i^T} + (1-\alpha)\frac{d}{2\theta_i^A}\right) + \left(\alpha\frac{d}{2\theta_j^T} + (1-\alpha)\frac{d}{2\theta_j^A}\right) \tag{47}$$

$$= \delta_{i*}^{weighted}(x) + \delta_{j*}^{weighted}(x) \tag{48}$$

$\square$

| Methods | ARC | GSM8k | Math. | Med. | MP. | Open. | SciQ | Social. | Truth. | Avg. |
|---|---|---|---|---|---|---|---|---|---|---|
| Qwen3-0.6B (Reward) | 92.67 | 93.54 | 93.27 | 73.50 | 79.44 | 91.63 | 96.28 | 74.26 | 81.08 | 86.19 |
| ReasoningRouter (Ours) | 94.01 | 95.78 | 94.60 | 74.06 | 82.20 | 92.88 | 97.52 | 80.74 | 81.98 | 88.20 |

*Table 2.* Comparison with reward model.

## C. Algorithm

We present the complete procedure for ReasoningRouter in Algorithm 1, which implements our routing strategy through four distinct phases.

---

**Algorithm 1** ReasoningRouter Inference Process

---

**Require:** Thinking embeddings $\{\boldsymbol{\lambda}_i^T\}_{i=1}^m$ and Answer embeddings $\{\boldsymbol{\lambda}_i^A\}_{i=1}^m$;
    Aggregation weights $w_T, w_A, w_C$; Triangulation parameters $\mathcal{A} = \{\alpha_1, \alpha_2\}$.
**Ensure:** Selected best model index $i^*$.
1: **// Phase 1: Marginal Quality Estimation via Triangulation**
2: **for** each $\alpha \in \mathcal{A}$ **do**
3:     **for** each pair of models $(i,j) \in \{1 \dots m\}^2$ **do**
4:         Compute combined distance: $\hat{\delta}_{ij}^{(\alpha)} \leftarrow \alpha \|\boldsymbol{\lambda}_i^T - \boldsymbol{\lambda}_j^T\|^2 + (1-\alpha)\|\boldsymbol{\lambda}_i^A - \boldsymbol{\lambda}_j^A\|^2$
5:     **end for**
6:     **for** each model $i = 1 \dots m$ **do**
7:         $\mathcal{S}_i \leftarrow \{\frac{1}{2}(\hat{\delta}_{ij}^{(\alpha)} + \hat{\delta}_{ik}^{(\alpha)} - \hat{\delta}_{jk}^{(\alpha)}) \mid j \neq i, k \neq i, j\}$
8:         $\hat{D}_i^{(\alpha)} \leftarrow \text{mean}(\mathcal{S}_i)$ {Estimate error variance}
9:     **end for**
10: **end for**
11: Solve Eq. (17) and (18) to obtain decoupled qualities $\hat{\theta}_i^T$ and $\hat{\theta}_i^A$ for all $i$.
12: **// Phase 2: Latent Truth Estimation**
13: $\hat{\mathbf{z}}^{*T} \leftarrow \left(\sum_{j=1}^m \hat{\theta}_j^T\right)^{-1} \sum_{j=1}^m \hat{\theta}_j^T \boldsymbol{\lambda}_j^T$ {Weighted consensus for Thinking}
14: $\hat{\mathbf{z}}^{*A} \leftarrow \left(\sum_{j=1}^m \hat{\theta}_j^A\right)^{-1} \sum_{j=1}^m \hat{\theta}_j^A \boldsymbol{\lambda}_j^A$ {Weighted consensus for Answer}
15: **// Phase 3: Causal Strength Estimation**
16: **for** each model $i = 1 \dots m$ **do**
17:     $\boldsymbol{\epsilon}_i^T \leftarrow \boldsymbol{\lambda}_i^T - \hat{\mathbf{z}}^{*T}$ and $\boldsymbol{\epsilon}_i^A \leftarrow \boldsymbol{\lambda}_i^A - \hat{\mathbf{z}}^{*A}$ {Compute residuals}
18:     $\hat{\beta}_i \leftarrow \frac{\langle \boldsymbol{\epsilon}_i^A, \boldsymbol{\epsilon}_i^T \rangle}{\|\boldsymbol{\epsilon}_i^T\|^2}$ {Causal coefficient via least squares}
19: **end for**
20: **// Phase 4: Scoring and Model Selection**
21: **for** each model $i = 1 \dots m$ **do**
22:     $\text{coherence}_i \leftarrow \frac{|\hat{\beta}_i|}{1+|\hat{\beta}_i|} \cdot \min(\hat{\theta}_i^T, \hat{\theta}_i^A)$ {Calculate causal coherence}
23:     $\hat{\theta}_i(x) \leftarrow w_T \hat{\theta}_i^T + w_A \hat{\theta}_i^A + w_C \cdot \text{coherence}_i$ {Final quality score}
24: **end for**
25: **return** $i^* = \arg\max_i \hat{\theta}_i(x)$

---

## D. Comparison with Reward Models

The remaining unsupervised category is reward-model-based routing (Lu et al., 2023), which uses another LLM to score output quality. We initially excluded this because it introduces a fundamentally different cost profile (full LLM inference for scoring vs. lightweight embedding extraction). Nevertheless, to address the reviewer's concern, we add a reward-model baseline using Qwen3-0.6B, a model of comparable scale to our embedding backbone, for the fairest possible comparison:

# E. Limitations and Scope

We discuss the main assumptions and scope of ReasoningRouter.

**Causal interpretation.**   ReasoningRouter uses the known generation order of large reasoning models to model a directed dependency from the thinking trace to the final answer. However, the estimated coefficient $\hat{\beta}$ should not be interpreted as a fully identified interventionist causal effect. Since thinking and answer embeddings are produced by the same model and encoder, shared input-dependent factors may confound their relationship. Our causal formulation is therefore best understood as dependency-aware modeling induced by temporal generation order, rather than strict causal identification.

**Embedding-space assumptions.**   The closed-form Causal Triangulation Property relies on simplified statistical assumptions, including approximately isotropic Gaussian noise and conditional independence of model errors around a latent task-specific target. Real LLM embeddings can be anisotropic and encoder-dependent, so these assumptions may only hold approximately. Our experiments across two embedding models suggest practical robustness, but extending the estimator to handle anisotropic covariance or correlated errors remains future work.

**Applicability to thinking-answer outputs.**   ReasoningRouter assumes that candidate outputs can be separated into a thinking component and an answer component. This matches many current reasoning models with explicit reasoning traces, but it is less directly applicable when reasoning and responses are not structured. Extending the framework to these settings may require task-specific segmentation or iterative dependency modeling.

