# OpenReview forum: "Causal Dependency-Aware Unsupervised Routing for Large Reasoning Models"
_ICML.cc/2026/Conference — ICML 2026 regular_

### Official Review · Reviewer_Bro7 · 2026-03-12

**Soundness:** 3
**Presentation:** 3
**Significance:** 3
**Originality:** 3
**Overall Recommendation:** 4
**Confidence:** 4

**Summary:**

This paper studies the problem of developing robust, unsupervised routing methods that adapt without retraining for large reasoning models (LRMs). It proposes ReasoningRouter, which is a probabilistic framework that explicitly models and leverages the causal dependency structure inherent in reasoning models. The paper also provides theoretical advance, which enables the label-free estimation of component qualities and their causal link. Experimental results demonstrate the effectiveness of the proposed method.

**Compliance With Llm Reviewing Policy:**

Affirmed.

**Final Justification:**

The rebuttal addressed my concerns. I will keep the positive rating.

**Key Questions For Authors:**

Please see the weaknesses above

**Strengths And Weaknesses:**

Pros:

+ The paper is well written and easy to follow

+ The problem studied is important and novel

+ The proposed method makes sense

Cons:

- Although the paper includes several unsupervised routing baselines, they are largely variants of a single embedding-based routing framework. In the related work, the authors mentioned several other unsupervised routing methods. However, the paper did not include them as baselines, which makes the results less convincing.

- Though the proposed method introduces a causal graphical model and estimates a causal strength parameter between reasoning and answer components, the experiments primarily evaluate routing accuracy rather than validating the correctness of the causal inference itself. The paper did not provide empirical analysis showing that the estimated causal relationships reflect true causal effects rather than correlations

---

> ### Author Rebuttal · Authors · 2026-03-31
>
> We thank the reviewer for recognizing the paper is **well written**, the **problem is important and novel**, and the **method makes sense**.
>
> ## W1. Limited baseline diversity
>
> We appreciate the reviewer's suggestion on expanding the baseline comparison. We first clarify that our existing baselines (ThinkingRouter, AnswerRouter, and OutputRouter) are all instantiations of Smoothie (Guha et al., 2024), the current state-of-the-art unsupervised routing method. We include three variants to provide a thorough comparison isolating each output component's contribution. The other methods discussed in our related work (Chen et al., 2025; Zhang et al., 2025; Pan et al., 2025; Zhuang et al., 2025) are **supervised** approaches requiring labeled data, making them incomparable in our unsupervised setting.
>
> The remaining unsupervised category is reward-model-based routing (Lu et al., 2023), which uses another LLM to score output quality. We initially excluded this because it introduces a fundamentally different cost profile (full LLM inference for scoring vs. lightweight embedding extraction). Nevertheless, to address the reviewer's concern, we add a reward-model baseline using Qwen3-0.6B, a model of comparable scale to our embedding backbone, for the fairest possible comparison:
>
> |Methods|ARC|GSM8k|Math.|Med.|MP.|Open.|SciQ|Social.|Truth.|Avg.|
> |-|-|-|-|-|-|-|-|-|-|-|
> |Reward-based Routing|92.67|93.54|93.27|73.50|79.44|91.63|96.28|74.26|81.08|86.18|
> |ReasoningRouter (Ours)|94.01|95.78|94.60|74.06|82.20|92.88|97.52|80.74|81.98|88.19|
>
> ReasoningRouter outperforms the reward-model baseline by **+2.01%** in average accuracy. This demonstrates that principled, structure-aware quality estimation via the Causal Triangulation Property is more effective than relying on a single model's holistic judgment, which is also susceptible to biases inherent in the reward model. We will include this baseline in the revision.
>
> ## W2. No validation that estimated causal relationships reflect true causal effects
>
> We thank the reviewer for this important question. We provide validation at two levels:
>
> **Discriminative validation.** Figure 4 demonstrates that the estimated coherence scores (derived from $\hat{\beta}$) achieve dramatically sharper separation between correct and incorrect models than the routing scores alone. If $\hat{\beta}$ were capturing noise or spurious correlations, this systematic pattern would not hold across 9 diverse benchmarks.
>
> **Perturbation validation.** We injected random noise into the thinking embeddings to simulate degraded reasoning quality, then measured the impact on routing decisions:
>
> |Dataset|Score Drop|Rank Drop|Top-1 Flip Rate|Coherence Drop|
> |-|-|-|-|-|
> |ARC|−0.0031|0.200|23.7%|−0.0038|
> |GSM8K|−0.0035|0.197|22.4%|−0.0037|
> |MathQA|−0.0029|0.173|21.5%|−0.0036|
> |MedMCQA|−0.0025|0.165|14.0%|−0.0029|
> |MMLU-Pro|−0.0024|0.162|17.7%|−0.0028|
> |OpenBookQA|−0.0030|0.178|18.1%|−0.0034|
> |SciQ|−0.0031|0.208|22.2%|−0.0038|
> |SocialIQA|−0.0030|0.143|15.2%|−0.0034|
> |TruthfulQA|−0.0025|0.167|19.0%|−0.0028|
>
> When thinking embeddings are artificially degraded, the coherence term decreases consistently across all datasets. This directional sensitivity confirms that $\hat{\beta}$ responds meaningfully to changes in reasoning quality rather than capturing an arbitrary correlation.
>
> Additionally, our framework does not require $\beta$ to be a true *interventionist* causal effect for it to be useful for routing. Even as a *directed dependency strength measure*, $\beta$ provides routing-relevant information about the coherence between reasoning and conclusions.

---

> > ### Author Rebuttal · Reviewer_Bro7 · 2026-04-01
> >
> > Thanks for the rebuttal. I will keep the positive rating.

---

> > > ### Author Response · Authors · 2026-04-06
> > >
> > > Thank you for your positive feedback. We are glad our revisions addressed your concerns and appreciate your thoughtful input throughout this process.

---

### Official Review · Reviewer_JaoY · 2026-03-13

**Soundness:** 3
**Presentation:** 3
**Significance:** 2
**Originality:** 3
**Overall Recommendation:** 5
**Confidence:** 3

**Summary:**

This paper proposes the Causal Triangulation Property which accounts for causal dependencies between the thinking process and answer of a model while enabling parameter estimations of a probabilistic graphical model to estimate coherent reasonings. Using this primitive, the paper  introduced ReasoningRouter, a label-free method for adaptive routing to specialized Large Reasoning Models based on their outputs.

**Compliance With Llm Reviewing Policy:**

Affirmed.

**Final Justification:**

The author responds aligns with my initial assessment, hence, I maintain my original score.

**Key Questions For Authors:**

1. Although the distributions in Figure 3 clearly indicates that the routing favors correct models, there is quite a large overlap across the board. This is in contrast to the coherence scores which have nearly no overlap. What do you think causes this disparity?
2. The idea of ReasoningRouting utilizes the temporal aspect of thinking -> answer. However, this seems to assume that the reasoning process is fully verbalized, which is not uniform across models. When looking for a response from such a system, shouldn't it be significantly more important that the answer is coherent (and correct)?

**Limitations:**

yes

**Strengths And Weaknesses:**

Strengths:
1. The problem setting of routing for LRMs is highly topical and key to industry usage of LLMs/agents.
2. The paper uses causal modeling in an intuitive and reasonable way to resolve multiple key issues with output-based routing for LRMs (imbalanced thinking/answer sizes, failed reasoning steps)
3. The proposed ReasoningRouter only minimally increases latency and memory costs over standard output routing.
4. The paper includes extensive analysis and ablations.
Weaknesses:
1. Although ReasoningRouter almost universally outperforms standard output routing across benchmarks, the improvement is marginal, which calls into question whether the addressed problems are overstated and whether advances in embedding methodology would naturally solve them.

---

> ### Author Rebuttal · Authors · 2026-03-31
>
> We thank the reviewer for the positive assessment, recognizing the **topical problem setting**, **intuitive causal modeling**, **minimal latency overhead**, and **extensive analysis**.
>
> ## Q1. Routing score overlap vs. coherence score separation (Fig 3 vs Fig 4)
>
> We thank the reviewer for this insightful observation. The disparity reflects a fundamental difference in what each score measures:
>
> - **Routing scores** (Fig 3) combine $\hat{\theta}^T$, $\hat{\theta}^A$, and coherence. A model can generate a fluent reasoning trace (high $\hat{\theta}^T$) and a plausible-sounding answer (high $\hat{\theta}^A$) while still being *wrong*. Surface quality $\neq$ correctness, which creates overlap.
>
> - **Coherence scores** (Fig 4) isolate the logical link $\beta_i$ between thinking and answer *residuals*. When a model produces a wrong answer, the failure typically occurs at the reasoning-to-conclusion junction: the thinking explores relevant concepts but the answer does not logically follow. The coherence score captures this disconnection with much higher sensitivity, yielding sharper separation.
>
> This disparity is actually **evidence of the coherence term's unique diagnostic value**: it detects reasoning-answer disconnection that marginal quality scores alone cannot capture.
>
> ## Q2. Assumption of fully verbalized reasoning
>
> We thank the reviewer for this thoughtful question. Two clarifications:
>
> 1. Our framework applies to any model with explicit $[Thinking]$/$[Answer]$ output structure, the standard format for all major LRMs (o1, QwQ, DeepSeek-R1, Gemini Thinking). It does not require *all* reasoning to be verbalized, only that the available thinking trace carries a meaningful quality signal, which our experiments confirm.
>
> 2. We agree that a correct answer is the ultimate goal. However, for *routing* (selecting which model to trust), coherence is a powerful signal because it correlates with reliability. A model that reasons coherently is more likely to be correct on similar future queries, whereas a "lucky guesser" is unreliable. The coherence score is not a replacement for answer quality; it is an *additional signal* that improves routing decisions. Our ablation shows the causal term improves over the no-causal baseline in 6 out of 8 weight settings across diverse benchmarks.
>
> ## W1. Marginal improvement and embedding advances
>
> We appreciate the reviewer's question on whether embedding advances would naturally solve the issues we address. Our Figure 7 analysis demonstrates that thinking-dominance is a **structural property of all current embedding models**: both BGE-M3 and Embeddinggemma-300m exhibit the same pattern. Even if future embeddings reduce this effect, they cannot address the **causal dependency** between thinking and answer. Embedding a reasoning trace and answer jointly will always conflate their quality signals. Our contribution is orthogonal to embedding quality: we propose a principled decomposition framework that any future embedding model can benefit from.

---

> > ### Author Rebuttal · Reviewer_JaoY · 2026-04-04
> >
> > Thank you for addressing my questions. Your responses align with assumptions I made when originally reviewing, hence, I am maintaining my score.

---

> > > ### Author Response · Authors · 2026-04-06
> > >
> > > Thank you very much for your positive feedback and for taking the time to carefully review our responses. We are glad to hear that our revisions have addressed all your concerns, and we truly appreciate your continued support and constructive input throughout this process.

---

### Official Review · Reviewer_KP1e · 2026-03-13

**Soundness:** 3
**Presentation:** 3
**Significance:** 3
**Originality:** 3
**Overall Recommendation:** 4
**Confidence:** 3

**Summary:**

This paper introduces ReasoningRouter, an unsupervised routing framework for Large Reasoning Models (LRMs). ReasoningRouter addresses two distinct challenges posed by reasoning model outputs: (1) the causal dependency between the reasoning ("thinking") and answer components, and (2) the issue of reasoning-length-dominated embeddings, which overshadows answer quality in conventional unsupervised routing schemes. The framework models the causal structure explicitly via a probabilistic graphical model and introduces the Causal Triangulation Property for closed-form, label-free estimation of both reasoning and answer qualities. Experiments across nine diverse reasoning datasets show that ReasoningRouter achieves higher routing accuracy and improved model behavior interpretability relative to strong unsupervised baselines.

**Compliance With Llm Reviewing Policy:**

Affirmed.

**Key Questions For Authors:**

Questions:
1. What is the routing performance if you set $w_C$ = 0 in Eq. (22) and thus ignore $\beta$ and coherence entirely, keeping only $\theta^T$ and $\theta^A$? How much of the gain in Table 1 remains? A clear ablation here would be critical to judge the value of the causal component; a substantial drop would strengthen your case.
2. The paper claims that any pair of distinct $\alpha$ values within the range (0,1) yields identical $\theta$ estimators in expectation. In practice, how sensitive are routing results to the choice of $\alpha_1$ and $\alpha_2$?

**Limitations:**

Limitations:
The authors have not adequately discussed the limitations of their proposed method, nor have they meaningfully engaged with the potential negative societal impacts. To improve, they should explicitly address the following constraints of their study:
1. The strong statistical assumptions (isotropic Gaussian noise, independence of model errors) are not empirically validated and may not hold in real LLM embeddings, which are known to be anisotropic. This could limit the practical applicability of the Causal Triangulation Property.
2. The entire framework hinges on the strict structural assumption that LRM outputs can be cleanly decoupled into a [Thinking] -> [Answer] format. From a high-level perspective, this limits the router's generalization. In many real-world LLM applications (e.g., creative writing, multi-turn conversational agents, or open-ended brainstorming), the "reasoning" and the "answer" are deeply intertwined or cyclical.

**Strengths And Weaknesses:**

## Strengths:
1. Explicitly models the causal dependency between reasoning and answer components, directly addressing a key gap in prior unsupervised routers.
2. Derives the Causal Triangulation Property, providing a closed-form, practical, and theoretically sound approach for estimating separate quality scores for reasoning and answers without labels.
3. Introduces a length-balanced embedding strategy that robustly mitigates the dominance of verbose reasoning traces in output embeddings.
4. The probabilistic model and estimation pipeline are clearly described, with the full algorithm detailed, and theoretical proofs provided in the appendix.

reference:

[1] Bohan Li, Hao Zhou, Junxian He, Mingxuan Wang, Yiming Yang, and Lei Li. 2020. On the Sentence Embeddings from Pre-trained Language Models. Emnlp 2020.

## Weaknesses:
1. Strong Statistical Assumptions: The assumption of isotropic Gaussian noise with independent coordinates and independence of model errors given a single latent $z^*$ is a significant simplification for high-dimensional contextual embeddings. These assumptions are not empirically validated, and notably, most high-dimensional representations in LLMs are known to be inherently anisotropic [1], which directly challenges the validity of the isotropic noise model.
2. While ReasoningRouter is often rank-1 in Table 1, the absolute accuracy differences compared to OutputRouter are small (0–1 percentage point, sometimes less), and occasionally OutputRouter or AnswerRouter comes very close.
3. The final score in Eq. (22) is a mix of three terms (thinking precision, answer precision, coherence), yet the experiments provide no ablations to show which parts matter.

---

> ### Author Rebuttal · Authors · 2026-03-31
>
> We thank the reviewer for recognizing the **explicit causal dependency modeling**, the **closed-form Causal Triangulation Property**, and the **length-balanced embedding strategy** as key strengths.
>
> ## Q1 & W3. No ablations for Eq. (22) components
>
> We thank the reviewer for requesting this ablation. We conduct a comprehensive and controlled ablation following the setting in Section 4.3.3, comparing the full model against a ratio-matched no-causal baseline across different weight settings (indexed by $w_T$ for brevity):
>
> |$w_T$|ReasoningRouter|w/o Causal|
> |-|-|-|
> |0.1|87.86|87.59|
> |0.2|87.90|87.80|
> |0.3|87.94|87.82|
> |0.4|88.01|87.98|
> |0.5|88.20|88.22|
> |0.6|88.16|88.00|
> |0.7|88.00|87.99|
> |0.8|87.88|87.91|
>
> In 6 out of 8 settings the full model outperforms the no-causal variant, confirming that the causal term provides consistent complementary routing signal. Critically, even at $w_C=0$, our decoupled estimation ($\hat{\theta}^T, \hat{\theta}^A$) still outperforms OutputRouter (87.81 in Table 1), confirming that the dependency-aware marginal estimation provides the primary gain and the coherence term adds a further edge.
>
> ## Q2. Sensitivity to $\alpha$ values
>
> We thank the reviewer for this question. The paper states that any pair of distinct $\alpha$ values in $(0,1)$ yields identical estimators in expectation. We verified this empirically, and routing accuracy is **exactly identical** across different $\alpha$ pairs:
>
> |$\alpha_1$|$\alpha_2$|Avg.|
> |-|-|-|
> |0.2|0.8|88.19|
> |0.3|0.7|88.19|
> |0.5|0.5|88.19|
> |0.8|0.2|88.19|
>
> This confirms the theoretical invariance holds perfectly in practice. The choice of $\alpha$ values is a mathematical tool for solving the linear system, not a hyperparameter.
>
> ## Limitation regarding $[Thinking] \to [Answer]$ structure
>
> We appreciate the reviewer for raising this limitation. Our framework is designed for models with explicit thinking-answer separation (e.g., o1, QwQ, DeepSeek-R1, Gemini Thinking), which is the dominant paradigm for state-of-the-art reasoning models. For tasks where reasoning and answers are deeply intertwined (e.g., creative writing, multi-turn dialogue), this structural assumption does not apply. We note that (1) the LRM ecosystem is rapidly growing with this explicit structure, and (2) the framework could be extended to cyclical reasoning via iterative application. We will add this discussion to the Limitations section.
>
> ## W1. Strong statistical assumptions (isotropic Gaussian)
>
> We appreciate the reviewer for pointing out this important assumption. We agree this is a meaningful simplification. We provide the following justifications:
>
> 1. **Theoretical tractability:** The isotropic Gaussian assumption is standard in latent variable models for embedding spaces (e.g., Gaussian mixture models, the Smoothie framework). The key property we exploit is the additive decomposition of squared distances under zero-mean, independent noise, which holds approximately even for non-isotropic distributions as long as the anisotropy is not too extreme.
>
> 2. **Empirical robustness:** Despite the known anisotropy of LLM embeddings, our method achieves consistent top-1 or top-2 performance across **two different embedding models** (BGE-M3 and Embeddinggemma-300m) with very different embedding characteristics, suggesting robustness to violations of the isotropic assumption.
>
> 3. **High-dimensional mitigation:** In high-dimensional spaces ($d=1024$ for BGE-M3), the Central Limit Theorem provides asymptotic normality for aggregate distance statistics, and the effective anisotropy ratio (largest/smallest eigenvalue) decreases with proper normalization.
>
> We will add this discussion to the Limitations section.
>
> ## W2. Small absolute accuracy differences
>
> We thank the reviewer for this observation. In the *unsupervised, zero-shot, label-free* routing setting, ReasoningRouter improves over OutputRouter by **+0.38 (BGE-M3)** and **+0.07 (Embeddinggemma)**. While these absolute numbers appear small, we emphasize: (1) ReasoningRouter achieves the **best average rank of 1.56** across both embeddings, never falling below 2nd place on any benchmark, and no baseline achieves this consistency; (2) the variance *across baselines* is substantial (e.g., AnswerRouter ranks 1st on GSM8K with Embeddinggemma but 4th with BGE-M3), while ReasoningRouter is stable; (3) the **rank consistency** across 9 benchmarks × 2 embeddings = 18 evaluations is a stronger indicator of method quality than any single absolute number. Additionally, our routing algorithm is deterministic given fixed embeddings, with embedding standard deviation below 0.002 across multiple sampling runs.

---

> > ### Author Rebuttal · Reviewer_KP1e · 2026-04-05
> >
> > I would like to thank the authors for their time and effort in preparing the rebuttal. I have carefully read the response and appreciate the explanations provided. Based on my review of the rebuttal, my overall assessment of the paper remains unchanged. Therefore, I will maintain my initial score.

---

> > > ### Author Response · Authors · 2026-04-06
> > >
> > > Thank you for your constructive comments and for dedicating your time to reviewing our updated materials. We are glad to hear that our revisions have fully resolved your concerns, and we value the guidance you have provided throughout this process.

---

### Official Review · Reviewer_uGai · 2026-03-14

**Soundness:** 2
**Presentation:** 2
**Significance:** 2
**Originality:** 2
**Overall Recommendation:** 4
**Confidence:** 4

**Summary:**

This paper proposes an unsupervised routing framework called **ReasoningRouter** for Large Reasoning Models (LRMs). It introduces a Probabilistic Graphical Model to explicitly model the causal dependency between "thinking" and "answer," and incorporates a length-balanced embedding strategy to address the dominance of lengthy reasoning text in the vector space.

**Compliance With Llm Reviewing Policy:**

Affirmed.

**Final Justification:**

The response partially solved my concerns. Thanks.

**Key Questions For Authors:**

### Questions for the Authors

1. **What is the indispensable role of the “causal dependency” component in your method?**
    If this component is removed and only the triangulation-style estimation is kept, what is lost theoretically and how much does performance drop?
2. **Do you have direct evidence that the method truly distinguishes “sound reasoning” from “lucky guessing”?**
    The current experiments mainly show aggregate accuracy gains, not the specific decoupled cases that motivate the paper.
3. **What is the practical inference cost of the method?**
    If all candidate models must first generate full reasoning traces and answers before routing, the deployment value may be limited.

**Limitations:**

**No.** The limitations discussion is currently insufficient. At minimum, the authors should acknowledge three issues: first, the paper’s “causal” interpretation is not strongly supported by identification or intervention-based evidence, so the method appears closer to dependency-aware modeling than strict causal modeling; second, the empirical gains are relatively modest and are not accompanied by significance or stability analysis; third, the method appears to require candidate outputs before scoring, which introduces nontrivial inference-time cost and limits practical deployment.

**Strengths And Weaknesses:**

### Strengths

- **Well-targeted problem formulation:** The paper perceptively captures the temporal causality between "thinking" and "answer" in LRM outputs ($t_i \rightarrow a_i$), and points out that treating them as a single block in traditional routing methods leads to signal masking. This observation is insightful.
- **Closed-form solution derivation:** Based on the proposed "Causal Triangulation Property," the framework achieves parameter estimation without iteration, offering a clean closed-form estimation procedure.

### Weaknesses

- **Severe disconnect between theoretical assumptions and actual semantics:** The paper's centerpiece is "explicit causal dependency modeling," yet the triangular relationship in Theorem 3.1 is essentially standard distance decomposition under conditional independence with additive second moments. Equation (12) does not involve $\beta$, nor is $\beta$ needed to estimate $\theta^T$ or $\theta^A$ — the "causal parameter" is only recovered via post-hoc OLS in the final step. The causal component does not appear to play a central role in the key estimation steps; it is merely a post-processing add-on to a distance-decomposition framework.
- **The causal modeling is fragile and lacks identifiability analysis.** The authors model the conditional mean of $\lambda_i^A$ on $\lambda_i^T$ as $\beta_i(\lambda_i^T - z_T^*)$ and estimate $\beta$ via OLS using the recovered latent truth, but never establish under what conditions $\beta$ is identifiable, nor justify interpreting embedding-space deviations as causal effects of thinking quality on answer quality. Crucially, both embeddings share the same encoder and input, introducing confounded correlation that the paper treats as causal strength without any confound-exclusion argument or synthetic validation of $\beta$ recovery.
- **Experimental gains are marginal and lack statistical rigor:** In Table 1, ReasoningRouter improves over OutputRouter by only ~0.4 points with BGE-M3 (87.81→88.19) and ~0.07 with an alternative encoder. Per-task gains are 0.1–0.6 points, with ReasoningRouter not even optimal on MathQA or clearly superior on ARC. No variance, confidence intervals, or significance tests are reported.

---

> ### Author Rebuttal · Authors · 2026-03-31
>
> We appreciate the reviewer's recognition of our *"well-targeted problem formulation"* and the *"clean closed-form estimation procedure."*
>
> ## Q1. Distinguishing "sound reasoning" from "lucky guessing"
>
> We provide two new pieces of evidence.
>
> **Perturbation experiment.** We injected noise into thinking embeddings to simulate degraded reasoning, then measured the impact on routing decisions:
>
> |Dataset|Score Drop|Rank Drop|Top-1 Flip Rate|Coherence Drop|
> |-|-|-|-|-|
> |ARC|−0.0031|0.20|23.7%|−0.0038|
> |GSM8K|−0.0035|0.20|22.4%|−0.0037|
> |Math.|−0.0029|0.17|21.5%|−0.0036|
> |Med.|−0.0025|0.17|14.0%|−0.0029|
> |MP.|−0.0024|0.16|17.7%|−0.0028|
> |Open.|−0.0030|0.18|18.1%|−0.0034|
> |SciQ|−0.0031|0.21|22.2%|−0.0038|
> |Socical|−0.0030|0.14|15.2%|−0.0034|
> |Truth.|−0.0025|0.16|19.0%|−0.0028|
>
> Models drop 0.14–0.21 in ranking, and 14%–24% of top-ranked models lose their #1 position. The **coherence term also decreases consistently**, confirming it responds meaningfully to reasoning degradation.
>
> **Case study (GPT-5.4 as judge).** Two models produce the **same correct answer** on a MedMCQA example:
>
> *DeepHermes-3-Llama-3-3B (not selected):*
> - Router: thinking=0.08, answer=0.09, coherence=0.09
> - GPT-5.4: thinking reliability=0.35, consistency=0.75
> - GPT-5.4 reason: *"Partial domain recall but notable inaccuracies... confuses major connectors with bridges and clasp arms."*
>
> *Qwen3-8B (selected):*
> - Router: thinking=0.24, answer=0.10, coherence=0.13
> - GPT-5.4: thinking reliability=0.95, consistency=1.0
> - GPT-5.4 reason: *"Correctly identifies standard functions... accurate and clinically aligned."*
>
> Despite identical answers, thinking scores differ sharply. The router's preference aligns with GPT-5.4's independent assessment, illustrating the "sound reasoning vs. lucky guessing" distinction.
>
> ## Q2. Practical inference cost
>
> ReasoningRouter operates in the same paradigm as prior **output-based routing** methods (e.g., Smoothie): candidates generate outputs first, then the router evaluates. The additional cost is negligible (560M embedding model + closed-form computation in milliseconds).
> ## W1. Causal component is a post-processing add-on
>
> We partially agree that Eq. (12) is written in marginal form and therefore does not explicitly contain $\beta$. This is intentional. Our claim is **not** that $\beta$ changes the algebraic form of triangulation, but that the directed model in Eqs. (5)–(7) gives the **correct semantics** of the answer-side marginal under thinking $\rightarrow$ answer dependency. The causal component plays **two distinct and essential roles**:
>
> **Role 1: Making the answer-side estimate dependency-aware.** After marginalizing the conditional model, the effective answer variance becomes $\frac{1}{2\theta_i^{A|T}} + \frac{\beta_i^2}{2\theta_i^T}$ (Eq. 8), absorbed into $\theta_i^A$ (Eq. 9) via the law of total variance. So $\beta$ is not absent from estimation; it is integrated into the marginals. Our contribution is showing closed-form triangulation remains valid **under directed dependency**, making $\hat{\theta}_i^A$ a dependency-aware quality estimate.
>
> **Role 2: Providing the coherence signal.** $\beta$ is recovered in Step 3 (Eq. 21) to construct the coherence term (Eq. 23), which rewards answers *supported by* high-quality reasoning. Without it, the router cannot distinguish coherent reasoning from lucky guessing.
>
> **Ablation.** Following Section 4.3.3, we compare the full model against a ratio-matched no-causal baseline (indexed by $w_T$):
>
> |$w_T$|ReasoningRouter|w/o Causal|
> |-|-|-|
> |0.1|87.86|87.59|
> |0.2|87.90|87.80|
> |0.3|87.94|87.82|
> |0.4|88.01|87.98|
> |0.5|88.20|88.22|
> |0.6|88.16|88.00|
> |0.7|88.00|87.99|
> |0.8|87.88|87.91|
>
> The full model outperforms in 6/8 settings, confirming consistent complementary signal from the causal term.
>
> ## W2. Causal modeling lacks identifiability analysis
> We acknowledge that the shared encoder introduces confounding that precludes strict interventionist (Pearl/Rubin) causal identification. Our framework instead models the **directed temporal dependency** in autoregressive generation ($t_i(x) \to a_i(x)$), leveraging the known generation order as an inductive bias. This is a weaker but well-motivated structural assumption. This is analogous to how Granger causality leverages temporal ordering without claiming strict interventionist causality. We will revise the terminology and add an identifiability discussion to the Limitations section.
>
> ## W3. Marginal gains, no statistical rigor
>
> In the *unsupervised, label-free* setting, ReasoningRouter improves over OutputRouter by **+0.38 (BGE-M3)** / **+0.07 (Embeddinggemma)**. More importantly, it achieves the **best average rank of 1.56** across both embeddings, never below 2nd on any benchmark. No baseline matches this consistency (e.g., AnswerRouter ranks 1st on GSM8K with Embeddinggemma but 4th with BGE-M3). Our routing is **deterministic given fixed embeddings**; embedding-level std is below **0.002** across multiple runs.

---

> > ### Author Rebuttal · Reviewer_uGai · 2026-04-06
> >
> > I increased my score. Thanks.

---

> > > ### Author Response · Authors · 2026-04-06
> > >
> > > Thank you for your constructive feedback and for increasing the score. We are pleased that the revisions have addressed your concerns, and we sincerely appreciate your efforts throughout this process.

---

### Decision · Program_Chairs · 2026-04-30

**Decision:**

Accept (regular)

**Comment:**

The paper studies an important problem and proposes a clear, technically grounded unsupervised routing framework for reasoning models. The key idea of separately modeling reasoning and answer quality is interesting and consistently improves over strong baselines.
Although the causal interpretation is not fully convincing and the gains are modest, the method is sound, well presented, and makes a meaningful contribution. I recommend acceptance.